# Metabarcoding the Bacterial Assemblages Associated with *Toxopneustes roseus* in the Mexican Central Pacific

**DOI:** 10.3390/microorganisms12061195

**Published:** 2024-06-13

**Authors:** Joicye Hernández-Zulueta, Sharix Rubio-Bueno, María del Pilar Zamora-Tavares, Ofelia Vargas-Ponce, Alma Paola Rodríguez-Troncoso, Fabián A. Rodríguez-Zaragoza

**Affiliations:** 1Departamento de Biología Celular y Molecular, Centro Universitario de Ciencias Biológicas y Agropecuarias (CUCBA), Universidad de Guadalajara, Zapopan 45200, Jalisco, Mexico; joicye.hernandez@academicos.udg.mx; 2Laboratorio de Ecología Molecular, Microbiología y Taxonomía (LEMITAX), Departamento de Ecología Aplicada, Centro Universitario de Ciencias Biológicas y Agropecuarias (CUCBA), Universidad de Guadalajara, Zapopan 45200, Jalisco, Mexico; 3Programa de Maestría en Ciencias en Biosistemática y Manejo de Recursos Naturales y Agrícolas, Centro Universitario de Ciencias Biológicas y Agropecuarias (CUCBA), Universidad de Guadalajara, Zapopan 45200, Jalisco, Mexico; sharixrubio@gmail.com; 4Laboratorio Nacional de Identificación y Caracterización Vegetal (LaniVeg), Departamento de Botánica y Zoología, Centro Universitario de Ciencias Biológicas y Agropecuarias (CUCBA), Universidad de Guadalajara, Zapopan 45200, Jalisco, Mexico; pilar.zamora@academicos.udg.mx (M.d.P.Z.-T.); ofelia.vargas@academicos.udg.mx (O.V.-P.); 5Laboratorio de Ecología Marina, Centro Universitario de la Costa (CUCosta), Universidad de Guadalajara, Puerto Vallarta 48280, Jalisco, Mexico; alma.rtroncoso@academicos.udg.mx

**Keywords:** microbial ecology, sea urchin microbiota, 16S rRNA, next-generation sequencing, bacteriome

## Abstract

The Mexican Central Pacific (MCP) region has discontinuous coral ecosystems with different protection and anthropogenic disturbance. Characterizing the bacterial assemblage associated with the sea urchin *Toxopneustes roseus* and its relationship with environmental variables will contribute to understanding the species’ physiology and ecology. We collected sea urchins from coral ecosystems at six sites in the MCP during the summer and winter for two consecutive years. The spatial scale represented the most important variation in the *T. roseus* bacteriome, particularly because of Isla Isabel National Park (PNII). Likewise, spatial differences correlated with habitat structure variables, mainly the sponge and live coral cover. The PNII exhibited highly diverse bacterial assemblages compared to other sites, characterized by families associated with diseases and environmental stress (*Saprospiraceae*, *Flammeovirgaceae*, and *Xanthobacteraceae*). The remaining five sites presented a constant spatiotemporal pattern, where the predominance of the *Campylobacteraceae* and *Helicobacteraceae* families was key to *T. roseus*’ holobiont. However, the dominance of certain bacterial families, such as *Enterobacteriaceae*, in the second analyzed year suggests that Punto B and Islas e islotes de Bahía Chamela Sanctuary were exposed to sewage contamination. Overall, our results improve the understanding of host-associated bacterial assemblages in specific time and space and their relationship with the environmental condition.

## 1. Introduction

Marine microorganisms are central drivers of ocean biogeochemical cycles; they control the emission of radioactive gases and are essential for community functioning because they provide ecosystem services [1,2]. The diversity of microbial metabolisms has allowed bacteria to succeed as free-living organisms and as symbionts [3]. These are relevant in multicellular organisms considered “holobionts” because they comprise the macroscopic host and its synergistic interdependence with bacteria, archaea, fungi, and other microorganisms [4,5].

The associations between marine invertebrates and microbial assemblages have been studied in recent years in terms of diversity, composition, and stability [6,7]. Specifically, the bacterial assemblage of marine invertebrates such as sponges and corals has been extensively characterized due to its ecological importance [8,9]. Sea urchins are considered a key group that maintains the function and stability of marine ecosystems. Because they control the abundance and distribution of other benthic species [10], sea urchins provide different microhabitats for a variety of bacteria that play a fundamental role in shaping and maintaining host homeostasis [11], including the establishment of native microbiota, digestion, nutrient cycling, health, and immunity [12].

Over the past 20 years, next-generation sequencing has improved our understanding of the diversity of symbiont bacteria and the benefits that bacteria offer to associated organisms, especially by providing nutrients and chemical defenses to their hosts [13]. These studies characterized the bacteria and compared different compartments of the digestive system of *Lytechinus variegatus* [11] and *Srongylocentrotus purpuratus* [14]. Changes in bacterial assemblage composition support the essential role of specific bacterial taxa on sea urchin health and digestion. Bacterial specificity depends on the type of feeding of sea urchins, such as the digestion of wood fragments and macroalgae [15,16]. Similarly, other studies have focused on the diversity and activity of sulfur bacteria, as well as sulfur metabolism, in the digestive tract of the sea urchin *Echinocardium cordatum* [17].

The Eastern Tropical Pacific includes the Mexican Central Pacific (MCP) region, which harbors important coral ecosystems that provide a favorable habitat for echinoderms [18]. In this region, the pink sea urchin *Toxopneustes roseus* (family *Toxopneustidae*) is one of shallow areas’ most notable and abundant echinoderms [19,20]. *T. roseus* has extensive ecological contributions, as it participates in energy transfer, conducts bioerosion processes, and helps maintain the integrity and persistence of algae in the ecosystem [21,22]. Like many sea urchins of the family Toxopneustidae (i.e., *T. pileolus* and *Tripneustes gratilla*), *T. roseus* contains a source of valuable metabolites such as lectins and bioactive substances with mitogenic and cytotoxic capacity [23,24,25]. Therefore, the sea urchin species *T. roseus* has biotechnological potential for drug development.

The ecological and biotechnological importance of *T. roseus*, its distribution along the MCP, and its proximity to human settlements make this species a study model to evaluate the response of the bacterial assemblage to diverse environmental conditions. Therefore, this study aimed to evaluate the spatiotemporal variation in the bacterial assemblage associated with *T. roseus* in the MCP and its relationship with several environmental variables. The results will contribute to identifying the variations in the structure of the bacterial assemblages associated with *T. roseus*. This knowledge improves our understanding of their richness, diversity, and community similarity changes in a specific time and space. Likewise, this knowledge will help establish some bacterial taxa as indicators of the general condition of the community with which they are associated.

## 2. Materials and Methods

### 2.1. Study Area

The MCP is located in an oceanographic area limited between two biogeographic provinces, the Warm Temperate Northeast Pacific and the Tropical Eastern Pacific [26]. This region has mixed conditions resulting from the confluence of currents of different origins [27,28,29]. Due to the geomorphological conditions of the region, the MCP is characterized by the discontinuous presence of reef ecosystems with different degrees of coral development, protection status, and human disturbance. The primary human disturbances include urban coastal development, fisheries, and tourism [30,31]. 

The study area covered insular and coastal sites within the MCP—Nayarit, Jalisco, and Colima (Figure 1)—with an approximate distance of 350 km. The sampling sites were as follows: (A) Isla Isabel National Park (21°50′39.62″ N, 105°52′46.96″ W), the northernmost site; (B) Islas Marietas National Park (20°41′54.1″ N, 105°34′58.1″ W), (C) Islas e islotes de Bahía Chamela Sanctuary (19°33′19.7″ N 105°06’28.8″ W), (D) Bahía Cuastecomates-Punta Melaque (19°13′51.7″ N 104°44′01.9″ W), and (E) Carrizales (19°05′44.9″ N 104°26′08.7″ W) in Bahía Ceníceros and (F) Punto B (19°5′55.21″ N, 104°23′24.47″ W) in Bahía Santiago, both on the coast of the southern boundary of the MCP. 

### 2.2. Sampling Collection

Samples were obtained in the warm–wet (summer) and cold–dry (winter) seasons of two consecutive years: summer 2017 (August–September), winter 2018 (January–February), summer 2018 (August–September), and winter 2019 (January–February). Three individuals of the sea urchin *T. roseus* with visual evidence of being healthy were collected for each sampling site and season, for a total of 72 samples. The organisms were transported to land, washed with sterile seawater in an aseptic space, and preserved in cryovials with liquid nitrogen. Each cryovial included a mixture of the pedicellaria, spines, digestive tract, and gonads of its respective replicate. Once in the laboratory, the cryovials were stored in ultracold conditions (−80 °C) until processing.

Variables representative of the habitat structure and seawater quality and conditions were determined for each sampling site and season (Appendix A). Sea surface temperature (SST), dissolved oxygen (DO), and salinity (PPM) were recorded in situ in triplicate with a YSI-556 multiparameter probe (Xylem Inc., Yellow Springs, OH, USA). Concentrations of chlorophyll α, phosphate (PO_4_), silicate (SIO_2_), nitrite + nitrate (NO_3_ + NO_2_), ammonium (NH4), total coliforms, and fecal coliforms were measured from seawater samples. Chlorophyll α concentration was estimated following Strickland and Parsons [32], while nutrients were determined following the Skalar San Plus II nutrient analyzer protocol. We determined seawater transparency with a Secchi disk to estimate the light extinction coefficient (turbidity) following the method proposed by Poole and Atkins [33]. 

The benthic habitat was recorded with video transects (25 m long) filmed 40 cm from the bottom. Each video transect was divided into 40 frames and 50 random systematic points (2000 points per video) to estimate the composition and coverage of benthic organisms and substrate types. Videos were analyzed with high-resolution monitors. Structural habitat elements were classified into hermatypic corals, hydrocorals, hydrozoans, octocorals, sponges, fleshy macroalgae, filamentous algae (turf), crustose coralline algae, articulated calcareous algae, and other sessile and vagile organisms. Coverages of other benthic components were designated as sandy substrate, rocky substrate, rubble, recently dead coral, and other benthic elements [34]. Depth (m) was recorded with dive computers, while topographic complexity was determined with the chain method [35]. 

### 2.3. Sequence Extraction and Analysis

Total DNA extraction was performed with the modified TRIzol Reagent protocol (Thermo Fisher Scientific, Life Technologies Corporation, Carlsbad, CA, USA) from the total sample of each cryovial (i.e., a mixture of the pedicellaria, spines, digestive tract, and gonads). The modifications included an additional Proteinase K treatment (20 μL) and a water bath incubation (65 °C) for each replicate. The extracted DNA (120 μL) was purified and individually concentrated with the GE Healthcare purification kit (Life Sciences™ illustra™ GFX™, Cytiva, Marlborough, MA, USA). Then, the DNA was quantified by fluorometry in Qubit^®^ 3.0 (Thermo Fisher Scientific).

Amplification of each replicate aggregate sample was performed by PCR using the modified 16S Metagenomics Kit (Thermo Fisher Scientific) and DreamTaq Green PCR Master Mix (2×) (Thermo Fisher Scientific). The two sets of primers included in the 16S Metagenomics Kit were used to amplify the following hypervariable regions of the bacterial 16S region: V2-4-8, and V3-6, 7-9. The final volume per reaction of the 16S Metagenomics Kit was 15 µL: 7.5 µL of Environmental Master Mix (2×), 1.5 µL of 16S Primer Set (10×), 3 µL of sample, and 3 µL of H_2_O. The final volume per reaction of the DreamTaq Green PCR Master Mix (2×) was 12.5 µL: 6.25 µL of DreamTaq Green PCR Master Mix (2×), 1.5 µL of 16S Primer Set (10×), 2 µL of sample, and 2.75 µL of H_2_O. Thermal cycler conditions were as follows: 10 min at 95 °C, 30 cycles of 30 s at 95 °C, 30 s at 58 °C, and 20 s at 72 °C, with a final step of 7 min at 72 °C. The presence and quality of amplicons were verified by electrophoresis on 1% agarose gels. 

Libraries were constructed using the Ion Plus Fragment Library Kit protocol. The ends were phosphorylated, adapters were ligated, and barcodes were added to each sample (50–100 ng of DNA) to identify the libraries, which were quantified with real-time PCR (qPCR) to determine their concentration and calculate the equimolar dilution factor for mixing. The template was prepared with an emulsion PCR on the Ion One Touch 2 System (Thermo Fisher Scientific, Life Technologies Corporation, Carlsbad, CA, USA). The concentration of the template was determined by fluorometry on the Qubit^®^ 3.0 kit (Thermo Fisher Scientific). The enriched template was loaded onto a PGM 318TM chip, following the instructions of the Ion PGM Hi Q View Sequencing Kit protocol for 400 bp fragments. Sequencing was performed on the Ion Torrent™ Personal Genome Machine^®^ (PGM) of the National Laboratory of Plant Identification and Characterization (University of Guadalajara).

The Ion Reporter™ 5.20.2.0 Software (Thermo Fisher, https://ionreporter.thermofisher.com/ir, full access date 27 May 2024) platform was used to detect and remove primers and short sequences (<150 base pairs) using the Metagenomics 16S w1.1 workflow (Thermo Fisher). Subsequently, the screened reads were clustered into operational taxonomic units (OTU) with 3% divergence, followed by removal of OTUs with less than ten copies. The final OTUs were taxonomically classified using the curated GreenGenes v13.5 [36] and MicroSEQ(R) 16S Reference Library v2013.1. Data analyses were performed at the family level because the libraries’ finest and most complete identification (99.3% of the sequences) was obtained under this classification. All sequences were uploaded to the National Center for Biotechnology Information repository under project number PRJNA742839.

### 2.4. Statistical Analysis

To evaluate the change in the structure and composition of the bacterial assemblage between years, seasons, and sites, we used a three-way experimental design with crossed factors (type I fixed-effect model) and no replication as a preliminary analysis. The season factor was removed from the design because it did not contribute to the explained variation of the model (Appendix A); in this manner, we used levels to increase replication in a simpler design. Therefore, the spatiotemporal variation in the bacterial assemblage associated with *T. roseus* was evaluated with a two-way experimental design with crossed factors and fixed effects (type I model), expressed as follows (Equation (1)):(1)$$Y=\mu +Year_{i}+Site_{j}+\left(Year_{i}\times Site_{j}\right)+\varepsilon_{ijl}$$
where *Y* was the multidimensional response variable or matrix; the *Year_i_* factor had two levels (periods 2017–2018 and 2018–2019); the *Site_j_* factor corresponded to six levels or sampling sites (Isla Isabel National Park, Islas Marietas National Park, Islas e islotes de Bahía Chamela Sanctuary, Bahía Cuastecomates-Punta Melaque, Carrizales, and Punto B); the term *Year_i_* × *Site_j_* expressed the interaction between the factors above, representing the spatiotemporal variation of the model; and *ε_ijl_* was the cumulative error of the model.

The sampling effort was evaluated with sample-based rarefactions to compare the expected bacterial family richness against the non-parametric estimators ICE, Chao 2, Jackknife 1, and Jackknife 2. Rarefaction curves were constructed as a global model for the entire sampling effort by considering replicates from all years, seasons, and sites. The curves were obtained with 10,000 random combinations without replacement with EstimateS V9.1 [37]. Then, they were plotted in SigmaPlot 12.0 (Systat Software Inc., San Jose, CA, USA).

The alpha diversity of the bacterial assemblage associated with *T. roseus* was analyzed with the estimation of total abundance (number of sequences) per family (N) and Hill numbers (^q^D): (a) family richness (^0^D); (b) Shannon’s exponential function of diversity (^1^D); (c) inverse of Simpson’s dominance (^2^D); and (d) Hill’s equity (^21^D) estimated as ^2^D/^1^D [38]. Indices were estimated per aggregate sample of replicates, and these were compared with a multidimensional, permutational analysis of variance (PERMANOVA) based on the experimental design, using data standardized to Z-values and a Euclidean distance matrix [39].

Beta diversity (composition and abundance of bacterial families) was evaluated with another PERMANOVA, using the same experimental design and statistical evaluation method but based on a matrix in Hellinger distances [40]. The contribution of bacterial taxa to the average dissimilarity between years and sites was estimated with a similarity percentage analysis (SIMPER) with Hellinger distances at a cut-off of 65% cumulative contribution to the average dissimilarity.

The variation in environmental variables was analyzed with a third PERMANOVA, using the experimental design described before but without replication. To do this, we used the data at the site level per year standardized to Z values, and we constructed a matrix in Euclidean distances [39]. Then, a SIMPER analysis was performed with Euclidean distances and a cut-off at 65% of the cumulative contribution of the environmental variables, using a two-way design crossed with factor Year and factor Site.

The results of the PERMANOVA for alpha and beta diversity were visualized in ordinates constructed with principal coordinate analysis (PCO). However, the third PERMANOVA (environmental variables) results were represented in a non-metric multidimensional scaling (nMDS). All ordinations were based on the pretreatments and similarity coefficients of their respective PERMANOVA. The alpha (Hill abundance and numbers) and beta (important bacterial families selected based on SIMPER results) diversity variables were represented as vectors based on multiple correlations. Spearman correlations were used for the environmental variables (relevant variables according to the SIMPER analysis). The overall statistical significance and that after the PERMANOVA tests was calculated with 10,000 permutations of residuals under a reduced model and type III sum of squares. When less than 100 permutations were obtained, Monte Carlo (MC) tests were used. Rarefaction curves, Hill number estimates, and ordinate analyses (PCO and nMDS) were developed in PRIMER v6.1.11 + PERMANOVA v1.01 [41].

The relationship between bacterial assemblages and environmental variables was evaluated at the sampling site level with a canonical redundancy analysis (RDA) [42]. The composition and abundance of the bacterial families previously treated with a Hellinger transformation were used as the biological matrix (Y), while the matrix of predictor variables (X) corresponded to the environmental variables previously standardized to Z-values (Appendix A). The multicollinearity of the predictor variables was reduced by considering those variables that presented a Pearson correlation (r) ≤ 0.75 and a variance inflation factor (VIF) < 10. The adjustment of the RDA model was calculated as the adjusted coefficient of determination (*R*^2^_adj_), based on the Trace statistic. The correlation among sites, bacterial families, and predictor variables was plotted in an ordination (triplot). In addition, a second RDA model was performed at the Site per Year level with the previous method (Appendix A). The results of the two-way SIMPER with crossed factors complemented this analysis. The statistical significance of both RDA models was determined with 10,000 Monte Carlo permutations under a reduced model in CANOCO v4.5 software [42].

## 3. Results

### 3.1. Characterization of the Bacterial Assembly of T. roseus 

The sequencing generated a total of 2,023,663 reads, of which 99.3% (2,009,347 reads) were identified at the family level. According to non-parametric estimators, the sampling effort ranged between 70.5 and 79.9%, representing 76.4% of the expected total richness of bacterial families (Appendix A). The 1855 OTUs obtained were classified into 190 families, 80 orders, 39 classes, and 19 phyla.

At the temporal level, the phylum Proteobacteria contributed the highest relative abundance (67.0–71.1%), followed by Bacteroidetes and Firmicutes (Figure 2A). In total, 109 families were shared between years, with 113 families recorded in the first year and 189 families in the second year of sampling. The dominant families were *Helicobacteraceae* (42.9–46.9%) and *Vibrionaceae* (6.4–7.1%) (Figure 2B). Other predominant families were *Burkholderiaceae* (4.1–5.3%), *Campylobacteraceae* (2.2–6.2%), *Flavobacteriaceae* (2.6–5.8%), and *Acholeplasmataceae* (2.1–3.9%).

From a spatial perspective, the six sites were characterized by the dominance of the phylum Proteobacteria, registering relative abundances between 38.6 and 91.1% (Figure 2C). Bacteroidetes (10.8–33.7%) was the second most abundant phylum, particularly at Punto B (PB), Islas Marietas National Park (PNIM), and Isla Isabel National Park (PNII). Firmicutes (7.0–10.3%) predominated in Bahía Cuastecomates-Punta Melaque (CUM) and Islas e islotes de Bahía Chamela Sanctuary (BCH), while in Carrizales (CRZ), the highest abundance was represented by the phylum Tenericutes (5.19%). Also, 23 families coincided among the six sites. A total of 181 families were recorded in PNII, 109 in PNIM, 55 in BCH, 61 in CUM, 59 in CRZ, and 43 in PB. In five of the six sites, the most abundant family was *Helicobacteraceae* (37.5–64.9%), whereas *Saprospiraceae* (16.1%) predominated at PNII (Figure 2D). The families *Vibrionaceae* (2.9–15.2%), *Flavobacteriaceae* (1.3–13.1%), *Desulfovibrionaceae* (0.8–15.2%), *Campylobacteraceae* (1.8–6.8%), and *Spiroplasmataceae* (1.8–13.9%) were also relevant due to their high abundances.

The bacterial families associated with *T. roseus* showed important changes in the PB and PNII sites (Appendix A) on a temporal scale. In the first year, PB recorded relatively high values of *Helicobacteraceae* (73.53%), changing in the second year by the predominance of *Desulfovibrionaceae* (21.55%) and *Flavobacteriaceae* (28.72%). A similar pattern was observed in PNII, where the *Flammeovirgaceae* (19.63%) and *Acholeplasmataceae* (15.58%) families represented the most abundant component in the first year, while *Saprospiraceae* (22.38%) and *Xanthobacteraceae* (8.64%) dominated in the second year.

### 3.2. Alpha and Beta Diversity

The PERMANOVA model of alpha diversity (N, ^0^D, ^1^D, ^2^D, ^21^D) explained 51.7% of the total variation; the Site factor was the only one that presented significant differences (37.3% of explained variation) (Table 1). The a posteriori tests showed that the spatial difference was given by the PNII with respect to the other sites, except for the PNIM (Appendix A); these two island sites are located in the northern limit of the study region. The PCO ordination of bacterial alpha diversity explained 98.1% of the total variation at the site level (Figure 3A). The first principal coordinate (PCO1) was associated with the highest ^0^D, ^1^D, and ^2^D values characteristic of the PNII. In contrast, the second principal coordinate (PCO2) was related to high N, found in the PNIM. The remaining sites, especially PB and BCH, correlated with ^21^D.

The PERMANOVA model of bacterial family composition and abundance explained 53.5% of the total variation (Table 1). The individual factors Year and Site presented significant differences, and spatial change was the most important (19.5% of the variation explained). During the second sampling year, the interannual variation was generated mostly by the increase in the abundances of the families *Helicobacteraceae*, *Vibrionaceae*, *Desulfovibrionaceae*, *Saprospiraceae*, *Xanthobacteraceae*, *Planococcaceae*, and *Spiroplasmataceae*. In addition, the abundance of the families *Flammeovirgaceae* and *Bartonellaceae* decreased. The results of the a posteriori tests of the Site factor were similar to those of alpha diversity, showing that the bacterial assemblage of the PNII was different from the other sites, except for PB (Appendix A). According to SIMPER analysis, the families with the highest contribution to average dissimilarity both in years and sites were *Helicobacteraceae*, *Desulfovibrionaceae*, *Burkholderiaceae*, *Flavobacteriaceae*, *Spiroplasmataceae*, and *Desulfonatronumaceae* (currently *Desulfonatronaceae*) (Appendix A). The families *Desulfovibrionaceae* and *Spiroplasmataceae* were only recorded in the second year (2018–2019), when *Helicobacteraceae*, *Flavobacteriaceae*, and *Desulfonatronaceae* showed higher relative abundance (Figure 2C). Similarly, the families *Helicobacteraceae*, *Desulfovibrionaceae*, and *Desulfonatronaceae* were representative at all sites except in PNII, whereas *Burkholderiaceae* and *Spiroplasmataceae* were found in low abundances at the PNII and PB sites. *Flavobacteriaceae* was present at most sites (except CUM), recording its highest relative abundance at PB (Figure 2D). The spatial patterns of the bacterial assemblage were observed on a PCO ordination that explained 75.7% of the total variation (Figure 3B). High abundance values of *Desulfonatronaceae* associated mainly with PB and BCH were observed at the first principal coordinate (PCO1). In contrast, *Helicobacteraceae* had higher representation at the PNIM, CRZ, and CUM sites in the second principal coordinate (PCO2).

The PERMANOVA model of the environmental variables explained 52.3% of the total variation (Table 1). The Site factor was the only one that presented significant differences, with 34.4% of the explained variation. However, subsequent tests did not allow for the determination of significant differences (*p* > 0.05; Appendix A). According to SIMPER analysis, the environmental variables with the greatest contribution to spatiotemporal differentiation were α-chlorophyll, dissolved oxygen, salinity, ammonium, fecal coliforms, and the cover of live coral, sponge, rock, and sand (Appendix A). The nMDS ordination showed a stress value of 0.15 (Figure 3C) and projected spatial distinctiveness of the PNII; this site positively correlated with a high sponge cover. Temporal differences in PB were attributed to fecal coliforms (Figure 3C).

### 3.3. Relationship of Bacterial Assemblage and Environmental Variables

Spatial RDA ordination was shown to be a significant model (*p*-value = 0.0437) and R^2^_adj_ = 62.0% (Figure 4A). The best combination of variables predictive of spatial change in the bacterial assemblage consisted of depth and live coral, sand, and sponge covers (Figure 4A and Appendix A). Sponge cover was the only significant individual environmental variable (Appendix A). The PNII was characterized by the highest sponge cover and average depth, which were positively related to higher abundances of *Desulfobacteraceae* and *Flammeovirgaceae* families and negatively related to the *Helicobacteraceae* family (Figure 4A). The highest live coral cover correlated with CUM and CRZ, followed by PNIM. These sites were mainly associated with the families *Spiroplasmataceae* and *Thermoactinomycetaceae*. In contrast, sites BCH and PB presented the highest sand cover. BCH was associated with high abundances of *Campylobacteraceae* and *Desulfonatronaceae*. PB correlated positively with *Enterobacteriaceae*, *Desulfovibrionaceae*, and *Flavobacteriaceae*; however, it correlated negatively with *Burkholderiaceae* and Clostridiales Family XI Incertae Sedis.

The spatiotemporal RDA resulted in a significant model (*p* = 0.0002) with an adjustment of R^2^_adj_ = 67.73% (Figure 4B). The best combination of predictor variables for this model consisted of environmental variables, such as sea surface temperature, salinity, dissolved oxygen, fecal coliforms, and light extinction coefficient, and benthic conditions, such as the presence of sponges, macroalgae, and sandy substrate (Figure 4B, Appendix A). Sponge cover, salinity, and dissolved oxygen were significant on their own (Appendix A). 

The spatiotemporal model complemented the results of the spatial RDA. In particular, these models highlight the spatial differentiation of PNII and the temporal differentiation of PB and BCH in the second year. PNII was characterized by the highest average sponge and macroalgae cover. In the first year, PNII correlated with the families *Methylobacteriaceae*, *Desulfobacteraceae*, and *Flammeovirgaceae*, among others. However, in the second year, it was positively associated with *Defluviitaleaceae* and *Saprospiraceae* (Figure 4B). In addition, the second year of sampling recorded higher average fecal coliform values in PB and BCH; this positively related to an increase in the *Desulfovibrionaceae*, *Desulfonatronaceae*, and *Enterobacteriaceae* families. In contrast, for both years, the other sites correlated positively with a high abundance of *Helicobacteraceae* and *Campylobacteraceae*, but negatively with high dissolved oxygen values, temperature, fecal coliforms, and sponge coverages.

## 4. Discussion

The results of this study evidenced that the structure and composition of the *T. roseus* bacteriome in the Mexican Central Pacific (MCP) showed certain spatial and temporal “stability” in most sites (PNIM, BCH, CUM, CRZ, and PB). These results agree with those reported by Hernández-Zulueta et al. [43], who found no spatial and temporal variation of the bacterial microbiota of *Pocillopora damicornis* and *P. verrucosa* in the MCP. These results suggest that bacterial assemblages are not influenced by physicochemical variables such as sea surface temperature, considering that the region presents daily variations of ±3 °C [44] and an annual temperature range of ±10 °C [45]. Littman et al. [46] and Carlos et al. [47] also observed this pattern in corals from the Great Barrier Reef in Australia and Brazil, respectively. These results are relevant because bacterial microbiota have been reported to contribute to marine invertebrates’ adaptation, resilience, and resistance to thermal stress generated largely by vents such as El Niño Southern Oscillation (ENSO) [48,49]. Therefore, the spatial stability of bacterial assemblages could be due to urchin microbiota specificity. This specificity might reflect the microbiota’s functions to maintain the host’s health, such as digestive health and innate immunity [12]. In addition, the lack of correlation between some environmental variables and the bacterial assemblage associated with *T. roseus* could be a result of the fact that the sampled individuals of this sea urchin belong to the same subpopulation, which is well adapted to the in situ environmental conditions despite the sites having different anthropogenic stress. Thus, *T. roseus* and its associated bacteria may be well adapted to local conditions. However, most bacteria associated with this sea urchin are non-culturable, so it is unknown which drivers (or variables) determine the spatiotemporal variation in their composition and abundance.

The only site that showed significant spatial and temporal change in the bacterial assemblage of *T. roseus* was Isla Isabel National Park (PNII). The PNII is an insular site located at the boundary of two biogeographic provinces [26], which promotes the richness and abundance of echinoderms representative of temperate and tropical zones [10,50]. Finally, the fact that it is an island site 28 km from the coast [43] could contribute to the differences in the *T. roseus* bacterial assemblages between the PNII and the rest of the sites sampled in the MCP.

Regarding the temporal differences in the *T. roseus* bacteriome in the PNII, a positive correlation was observed in the *Flammeovirgaceae* family with the macroalgal coverage of the PNII in the first year analyzed (2017–2018). Increases in this family of opportunistic bacteria have been associated with prolonged thermic stress [51] and disease in asteroids [52]. However, this pattern changed the following year (2018–2019), with the predominance of the *Saprospiraceae* and *Xanthobacteraceae* families. The *Saprospiraceae* family has been associated with bald disease in sea urchins [13], while an abundance of *Xanthobacteraceae* has been found in bleached algae [53]. Therefore, our results suggest that the high values of *Flammeovirgaceae* and *Xanthobacteracea* families in PNII might be remnants of the 2014/2016 ENSO event [54].

With the exception of the PNII, the *T. roseus* bacteriome maintained a similar pattern across the years studied (2017–2018 and 2018–2019), with no differences associated with local conditions [55,56]. These findings support the hypothesis of spatial and temporal adaptability of species to their environment [57]. This adaptability might be the result of a coevolutionary process [58] or a reflection of the host’s ability to structure its bacterial microbiota, modifying the abundance of specific taxa obtained from the environment, regardless of site-linked variations [55]. In this sense, the environmental changes that occur throughout the year in most studied sites may be within the tolerance ranges that allow *T. roseus* and its associated bacterial microbiota to survive and grow.

The prevalence of Proteobacteria and Bacteroidetes primarily characterized the bacteriome. These phyla are the most abundant members of the bacterial microbiota of echinoids [59,60,61]. In the present study, they represented the major component of the bacterial assemblage of *T. roseus*, collectively contributing to 43.6–68% of the bacterial relative abundance of the sea urchin. Particularly, these families dominate in the coelomic fluid, feces, and intestinal tissue of some echinoid species [12,14,62], and they indicate the mutualistic relationship between the sea urchin and its resident microbiota [11] to carry out digestion and nutrient uptake processes [12].

Furthermore, the echinoderm assemblage in the MCP is strongly related to local conditions, particularly to structural elements of the benthic habitat [63]. Therefore, variables representative of habitat structure explained the spatial change of the bacterial assemblage of *T. roseus*. Live coral cover (LCC) was an important predictor variable for *T. roseus* bacterial assemblage. CRZ had the highest average LCC (67.6%) of all sites, while BCH, CUM, and IMNP presented values between 16 and 21%. Hernández-Zulueta et al. [64] observed that the environmental variables that explain the variation in bacterial assemblages of mucus and tissue of *P. damicornis* and *P. verrucosa*, were the coverages of fleshy macroalgae, live coral, and sponges. These results suggest that the coral bacterial assemblage is interconnected with the bacterial microbiota of other structural elements of the benthos.

Due to the close relationship of sea urchins with the bare sediment, the sandy substrate represented a key component of the bacterial assemblage of echinoids. The bacterial microbiota of marine sediment are highly diverse and different from other organisms [59,60]. Thus, the sediment acts as a reservoir of microorganisms that the holobiont can obtain and filter, selecting specific taxa according to the internal conditions of the sea urchin [55]. However, our study did not determine the bacterial microbiota of the bare sediments surrounding the *T. roseus* (which would have served as a control because they possibly reflect anthropogenic pollution). This should be considered in future studies to provide more information about the conditions of the studied sites.

Punto B, the site with the least coral development (LCC < 5%) and the greatest anthropogenic impact, showed the highest abundance of the *Enterobacteriaceae* family, which has been associated with fecal coliforms [65,66]. In addition, this site is in front of a large urban development and receives constant wastewater discharges [43], leading to changes in bacterial assemblages [65,66]. Punto B also showed a dominance of *Prolixibacteraceae* and *Sphingobacteriaceae* during the second year of sampling. *Prolixibacteraceae* has been reported as the second most abundant family in wetlands of contaminated sites due to its proximity to coastal urban developments [67], while *Sphingobacteriaceae* includes members resistant to physical disturbance and heavy metals [68,69,70]. In addition, *Flavobacteriaceae* and *Desulfovibrionaceae* presented high relative values in PB and BCH from 2018 to 2019. Although *Flavobacteriaceae* is found in low amounts in urchins [11,13], this family predominates in sea urchins living near urban developments [62]. Similarly, other marine invertebrates, e.g., corals, with a disease or some level of environmental stress show an overrepresentation of some bacterial taxa, including Flavobacteriales and other members of *Desulfovibrionaceae* [71,72].

This study presents the first spatiotemporal characterization of the bacterial assemblage associated with the sea urchin *T. roseus* in the MCP. Our results indicate that the changes in environmental conditions derived from seasonality do not represent a significant factor influencing the *T. roseus* bacteriome at the regional level. The structure, composition, and abundance of the *T. roseus* bacterial assemblage at most sites (PNIM, BCH, CUM, CRZ, and PB) suggested the plasticity of the holobiont in the face of spatial and temporal environmental variability in the MCP. In addition, the dominance of some families of the order Campylobacterales (i.e., *Helicobacteraceae* and *Campylobacteraceae*) is critical because of their possible relationship with urchin health at the physiological level. However, the environmental differences of the PNII site suggest that the bacterial microbiota of the pink sea urchin *T. roseus* may be dynamic and present important changes in its composition and abundance.

Echinoid diseases have not yet been reported in the MCP. However, the particular prevalence of some bacterial taxa implicated in diseased or stressed marine organisms (i.e., *Flavobacteriaceae*, *Sphingobacteriaceae*, *Saprospiraceae*, and *Flammeovirgaceae*, among others), suggests that some populations of *T. roseus* are under stress, mainly in the PB and PNII sites. In addition, the low representation of the *Helicobacteraceae* and *Campylobacteraceae* families in the PB and PNII sites could indicate unhealthy hosts. The dominance of *Enterobacteriaceae* suggests that the PB and BCH coral ecosystems (2018–2019) were exposed to possible anthropogenic contamination.

Finally, this work showed that reef habitat structure significantly influences the local bacterial assemblage of *T. roseus*. The relationship we found between high live coral cover (LCC) and sites with similar assemblages emphasizes the importance of continuing efforts to protect and conserve the main coral ecosystems in the MCP.

## Figures and Tables

**Figure 1 microorganisms-12-01195-f001:**
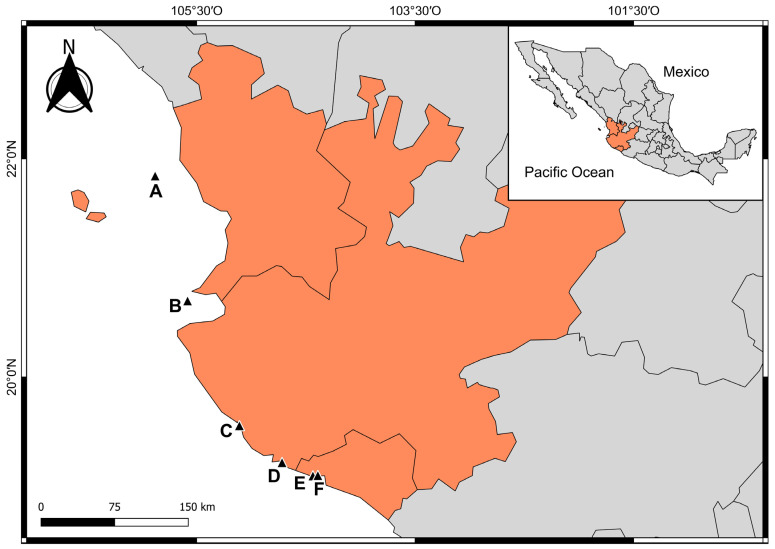
Study area in the Mexican Central Pacific. Sites: (A) Isla Isabel National Park (PNII) Nayarit, (B) Islas Marietas National Park (PNIM), Nayarit, (C) Islas e islotes de Bahía Chamela Sanctuary (BCH), Jalisco, (D) Bahía Cuastecomates-Punta Melaque (CUM), Jalisco, (E) Carrizales (CRZ), Colima, (F) Punto B (PB), Colima. The orange area represents the Nayarit, Jalisco, and Colima states.

**Figure 2 microorganisms-12-01195-f002:**
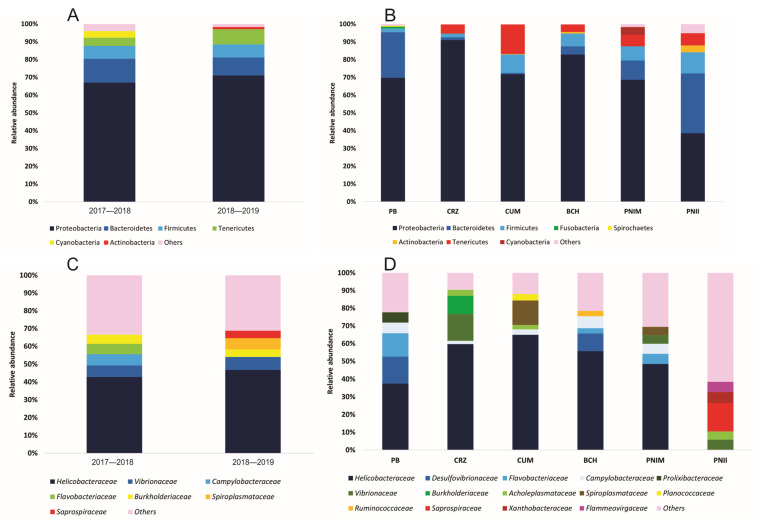
Average relative abundance of the most abundant bacterial taxa of the sea urchin *T. roseus*. The taxonomic classification is expressed at the level of phyla by year (**A**) and site (**B**), as well as families by year (**C**) and site (**D**). Codes: PNII: Isla Isabel National Park; PNIM: Islas Marietas National Park; BCH: Islas e islotes de Bahía Chamela Sanctuary; CUM: Bahía Cuastecomates-Punta Melaque; CRZ: Carrizales; PB: Punto B.

**Figure 3 microorganisms-12-01195-f003:**
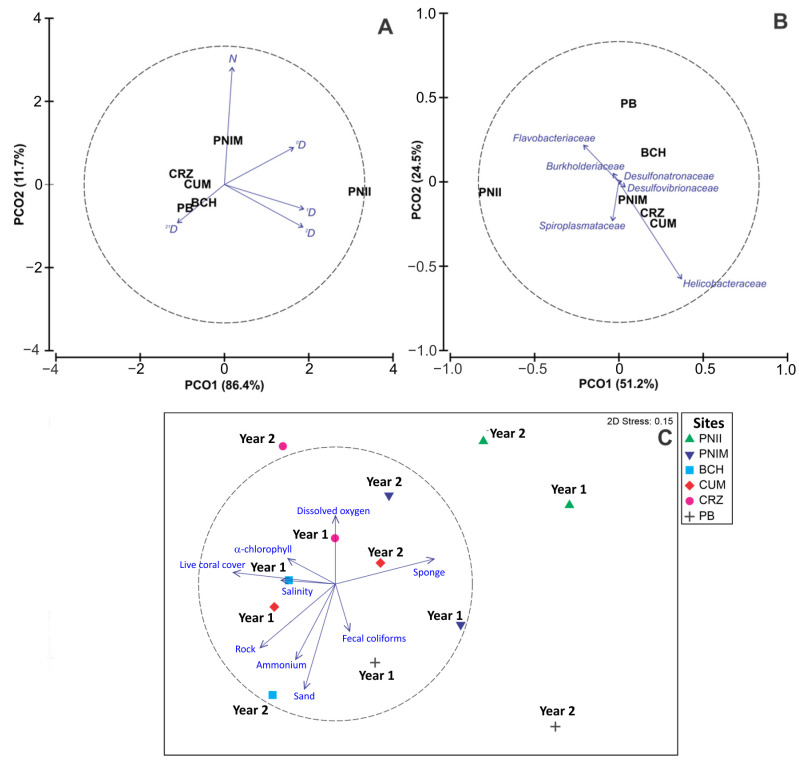
Principal coordinate analysis (PCO) ordinations of (**A**) alpha diversity (N, ^0^D, ^1^D, ^2^D, ^21^D); (**B**) composition and abundance of the families (blue arrows) of the bacterial assemblage associated with the sea urchin *T. roseus* per site. (**C**) Non-metric multidimensional scaling (nMDS) ordination of environmental variables at the site level by year. Codes: PNII: Isla Isabel National Park; PNIM: Islas Marietas National Park; BCH: Islas e islotes de Bahía Chamela Sanctuary; CUM: Bahía Cuastecomates-Punta Melaque; CRZ: Carrizales; PB: Punto B; Year 1: 2017–2018; Year 2: 2018–2019.

**Figure 4 microorganisms-12-01195-f004:**
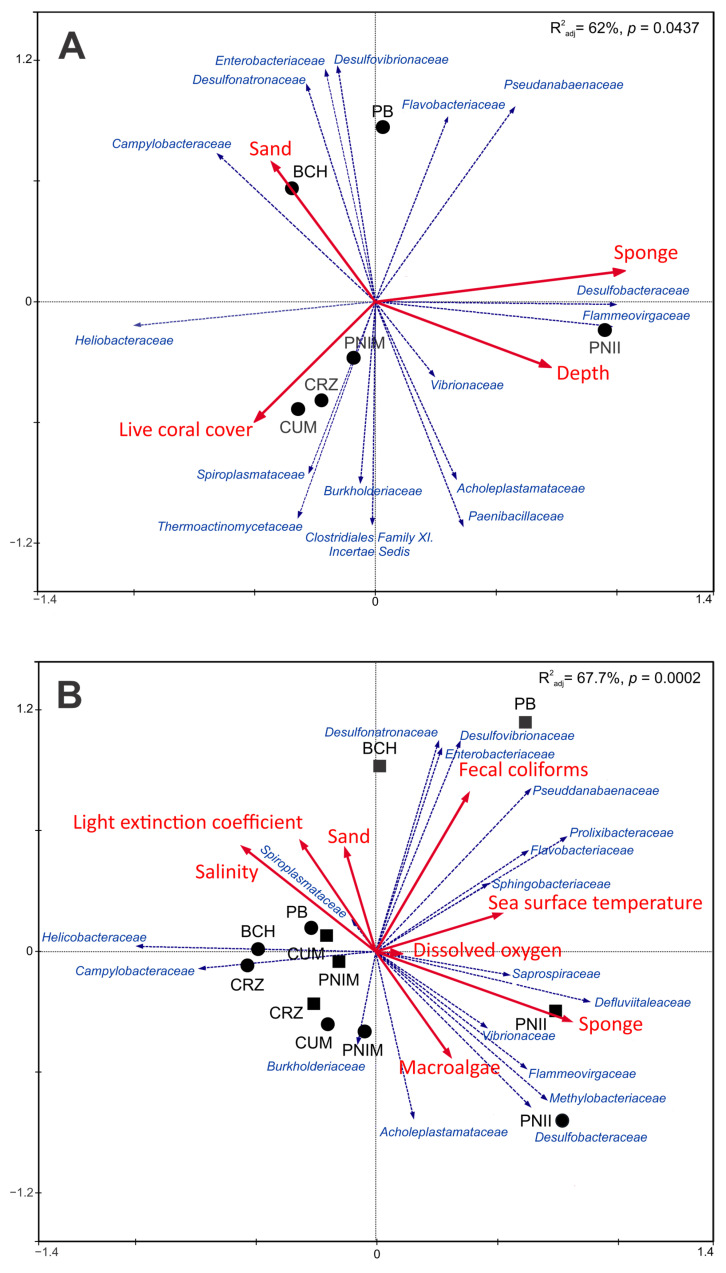
Canonical redundancy analysis (RDA) ordinations of the spatial and spatiotemporal variation in the bacterial assemblage associated with the sea urchin *T. roseus* and its relationship with environmental variables in the MCP. Bacterial families (represented with blue arrows) were selected based on the results of RDA and SIMPER outputs (see Appendix A). Predictor variables are shown with red arrows. In (**A**), the black circles correspond only to the sites, while in (**B**), the black circles and squares correspond to the sites sampled in the first and second years, respectively. Codes: PNII: Isla Isabel National Park; PNIM: Islas Marietas National Park; BCH: Islas e Islotes de la Bahía de Chamela Sanctuary; CUM: Bahía Cuastecomates-Punta Melaque; CRZ: Carrizales; PB: Point B; Year 1: 2017–2018 (black circles); Year 2: 2018–2019 (black squares).

**Table 1 microorganisms-12-01195-t001:** Results of the two-way PERMANOVA with crossed factors for the alpha (N, ^0^D, ^1^D, ^2^D, ^21^D) and beta diversity (composition and abundance of families) of the bacterial assemblage associated with the sea urchin *T. roseus* and for the environmental variables. Codes: C.V.% = Coefficient of explained variation in percentage. Values in bold correspond to significant differences (*p* ≤ 0.05).

Factors	Pseudo-*F*	*p*	C.V.%
Alpha diversity			
Year	2.0658	0.1336	14.4
Site	3.3813	**0.0127**	37.3
Year × Site	0.34989	0.9788	0.0
Residuals			48.3
Beta diversity			
Year	0.89	**0.0283**	16.1
Site	0.62	**0.0277**	19.5
Year × Site	0.47	0.1463	17.9
Residuals	0.36		46.5
Environmental Variables			
Year	1.8421	0.1162	17.9
Site	2.0416	**0.0032**	34.4
Residuals			47.7

## Data Availability

The raw data supporting the conclusions of this article will be made available by the authors on request.

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
