# Peer review of "Metabarcoding the Bacterial Assemblages Associated with Toxopneustes roseus in the Mexican Central Pacific"

_microorganisms, 2024, doi:10.3390/microorganisms12061195_

Round 1
Reviewer 1 Report
Comments and Suggestions for Authors
The article needs minor revisions, since it includes a lot of data obtained using adequate methods, as well as comprehensive introduction and discussion.
1. Line 114. “Each cryovial included the pedicellaria, spines, digestive tract, and gonads of its respective replicate.” Did you use total sample to isolate DNA?
2. Table S1. The table is unclear. Table S1 seems to show a list of parameters used for water analysis and parameters used for different models. In the same time, differences between table columns are not clear. Please, add short explanation to the manuscript or/and explanation in footnote.
3. Figs. 2A and 2B demonstrate results obtained at different years, while the samples were collected during different seasons. Could you modify these figures to show difference between the seasons.
4. Line 308. Italicize family names (and check the italics in the text).
5. The result description and discussion are focused on the diversity at family level. In the same time, each family may include genera with different properties and environmental role. Thus, could you add description of main genera detected at different samples.
Author Response
Response to reviewers
REVIEWER 1.
The article needs minor revisions, since it includes a lot of data obtained using adequatemethods, as well as comprehensive introduction and discussion. 1. Line 114. “Each cryovial included the pedicellaria, spines, digestive tract, and gonads ofits respective replicate.” Did you use total sample to isolate DNA?Authors response:
Answer 1. The authors appreciate all the time the reviewer took to improve the quality of the manuscript. The total DNA was isolated; to avoid confusion, the manuscript was modified as follows: The mixture of pedicellaria, spines, digestive tract, and gonads was used to extract total DNA (see lines 114 and 138-140).
2. Table S1. The table is unclear. Table S1 seems to show a list of parameters used forwater analysis and parameters used for different models. In the same time, differencesbetween table columns are not clear. Please, add short explanation to the manuscriptor/and explanation in footnote.
Answer 2. Thanks a lot for your comment. Table S1 was modified to present the information on environmental variables used in the RDA models in a clearer way. For this purpose, Table S1 was edited, and footnotes were included to explain how these variables were selected (see Table S1).
3. Figs. 2A and 2B demonstrate results obtained at different years, while the samples werecollected during different seasons. Could you modify these figures to show differencebetween the seasons.
Answer 3. Thank you very much for your comment. Figure 2 was not modified as the season factor is not longer included in the experimental design. Previous statistical analyses showed that variation by season resulted as not significant (p > 0.05), and therefore discarded. This new analytical approach increased replication at the spatial (analysis between sites) and temporal levels (analysis between years) (see Table S2 and lines 180-183).
4. Line 308. Italicize family names (and check the italics in the text).
Answer 4. We apologize for the mistake, and in the new version of the manuscript all the family names are italicized; this was also applied in Tables and Figures section.
5. The result description and discussion are focused on the diversity at family level. In thesame time, each family may include genera with different properties and environmentalrole. Thus, could you add description of main genera detected at different samples.
Answer 5. Thank you very much for your comment. In our study, we worked at the family level because NGS platforms generated short read lengths of approximately 250 to 400 bp that provide poor phylogenetic information compared to full-length 16S rRNA gene sequences (∼1500 bp) (see Glenn, 2011 and Bailen et al., 2020). In addition, taxonomic classification was done with the curated GreenGenes v3.13.5 database, in which taxonomic resolution at the genus level could not be obtained because most OTUs were only related to the family level.
Bailén M, Bressa C, Larrosa M, González-Soltero R. 2020. Bioinformatic strategies to address limitations of 16rRNA short-read amplicons from different sequencing platforms. J Microbiol Methods. 169:105811. doi: 10.1016/j.mimet.2019.105811.
Glenn TC (2011) Field guide to next-generation DNA sequencers. Mol Ecol Resour 11: 759–769.

Reviewer 2 Report
Comments and Suggestions for Authors
The paper by Rubio-Bueno et al describes the microbiome of sea urchins in different sites on the Mexican Central Pacific and investigates host-microbiome patterns and how are affected by spatio-temporal environmental changes. Overall, the paper is well written and I will it endorse for publication.
Below some comments that i feel will help the manuscript.
The authors collected sea urchins that are in proximity to the sediments; yet they have not collected a bare sediments from the same sites as controls for the in situ sediment microbiota on their sites. Examining the in situ sediment microbiota in parallel with their samples would have enhanced their argument that certain sites are under anthropogenic contamination. Pollution is there (i do not doubt that), and therefore it should be also reflected on the bare sediment, solidifying your argument that affects also the sea urchins and other invertebrates. Since these bare sediments were not collected (unless if i missed it), I would suggest to link what you write in paragraph starting on line 467 with the fact that "although your study did not include analyzing a bare sediment along with your samples as sediment microbiota control, you know that is typical for the the XX and YY sites (PB and BCH) sites to receive contamination that is reflected on the sea uchins."
You write on line 440: These findings support the hypothesis of spatial and temporal adaptability of species to their environment [57]. This adaptability might be the result of a coevolutionary process [58] or the reflection of the host’s ability to structure its bacterial microbiota, modifying the abundance of specific taxa obtained from the environment, regardless of site-linked variations [55].
I think that here you also need to acknowledge, that aside from the co-evolution and adaptability, the environmental changes that occur throughout the year in some sites are within the tolerance range that allow the host, and subsequently the symbionts, (microbiome) to survive and grow.
Similarly, I think you should stress more that although in some sites you find no correlations with environmental features e.g., with ToC, yet you have instances where you are referring to thermal stress-related microbiomes. So it feels like that you need to write a sentence or two and explain to the reader that you might have sampled sub-populations that are well (or not), adapted to the in situ conditions. This can be either because the ecosystems that you sampled are under human-related stress and therefore your samples struggle/are not yet well adjusted to these changes, or that these changes are higher than the tolerance range of the sea urchins or something towards this direction.
Author Response
Response to reviewers
REVIEWER #2
The paper by Rubio-Bueno et al describes the microbiome of sea urchins in different sites
on the Mexican Central Pacific and investigates host-microbiome patterns and how are
affected by spatio-temporal environmental changes. Overall, the paper is well written and I
will it endorse for publication.
Below some comments that i feel will help the manuscript.
1. The authors collected sea urchins that are in proximity to the sediments; yet they have not collected a bare sediments from the same sites as controls for the in situ sediment microbiota on their sites. Examining the in situ sediment microbiota in parallel with their samples would have enhanced their argument that certain sites are under anthropogenic contamination. Pollution is there (i do not doubt that), and therefore it should be also reflected on the bare sediment, solidifying your argument that affects also the sea urchins and other invertebrates. Since these bare sediments were not collected (unless if i missed it), I would suggest to link what you write in paragraph starting on line 467 with the fact that although your study did not include analyzing a bare sediment along with your samples as sediment microbiota control, you know that is typical for the the XX and YY sites (PB and BCH) sites to receive contamination that is reflected on the sea uchins.
Answer 1. Thank you very much for your comment, and to include this idea and avoid confussions, the manuscript was modified as follows: “However, our study did not determine the bacterial microbiota of the bare sediments surrounding T. roseus (which would have served as a control because they possibly reflect the anthropogenic pollution). This should be considered in future studies to provide more information about the conditions of the studied sites” (lines 481-484).
2. You write on line 440: These findings support the hypothesis of spatial and temporal adaptability of species to their environment [57]. This adaptability might be the result of a coevolutionary process [58] or the reflection of the host’sability to structure its bacterial microbiota, modifying the abundance of specific taxa obtained from the environment, regardless of site-linked variations [55].
I think that here you also need to acknowledge, that aside from the co-evolution and adaptability, the environmental changes that occur throughout the year in some sites are within the tolerance range that allow the host, and subsequently the symbionts, (microbiome) to survive and grow.
Answer 2. Thanks for this comment. We included the following sentence: “In this sense, the environmental changes that occur throughout the year in most studied sites may be within the tolerance ranges that allow T. roseus and its associated bacterial microbiota to survive and grow” (lines 455-457).
3. Similarly, I think you should stress more that although in some sites you find no correlations with environmental features e.g., with ToC, yet you have instances where you are referring to thermal stress-related microbiomes. So it feels like that you need to write a sentence or two and explain to the reader that you might have sampled sub-populations that are well (or not), adapted to the in situ conditions. This can be either because the ecosystems that you sampled are under human-related stress and therefore your samples struggle/are not yet well adjusted to these changes, or that these changes are higher than the tolerance range of the sea urchins or something towards this direction.
Answer 3. Thank you very much for your comment. We included the following sentence: “On the other hand, the lack of correlation between some environmental variables and the bacterial assemblage associated with T. roseus could be a result of the fact that the sampled individuals of this sea urchin belong to the same subpopulation, which is well adapted to the in situ environmental conditions despite the sites have different anthropogenic stress. Thus, it is possible T. roseus and its associated bacteria are well-adapted to local conditions. However, most of the bacteria associated with this sea urchin are non-culturable, so it is unknown which drivers (or variables) determine the spatio-temporal variation in their composition and abundance” (lines 423-431).
